# The Entrepreneurship Ecosystem of Food Festivals —A Vendors' Approach

Luiza Ossowska, Dorota Janiszewska  and Grzegorz Kwiatkowski *

Department of Economics, Koszalin University of Technology, 75-343 Koszalin, Poland
* Correspondence: grzegorz.kwiatkowski@tu.koszalin.pl or gregory.kwiatkowski@hvl.no

**Abstract:** A food festival is a type of an environment in which various stakeholders function. The main aim of the research is to indicate the elements and relations of the entrepreneurship ecosystem of food festivals. Empirical data were collected at three food festivals in Poland in the summer of 2020 using the pen-and-paper interview method and semi-structured interviews. During the research, a total of 58 interviews were conducted with vendors. A coding technique was used to process the data. The themes included in the interviews concerned parts of the food festival entrepreneurship model: capital, micro-environment, and macro-environment. The conducted research shows that the core of the entrepreneurship ecosystem model of food festivals is dominated by the family capital. For the vendor, the food festival acts as a platform connecting with the micro-environment and other stakeholders. In the macro-environment of the food festival ecosystem, apart from the conditions of support, there are also factors that limited the activity. The research is an attempt to fill the gaps in the identification of specific features and elements of the entrepreneurial ecosystem of food festivals. The research is an attempt to show how the entrepreneurship ecosystem model of food festivals works. The observations require further in-depth research, e.g., in terms of the evolution of this ecosystem or the dynamics of relationships.

**Keywords:** food festival; food producer; food vendor; entrepreneurship ecosystem

## 1. Introduction

Food festivals are defined as events focusing on food products. These are planned, organized, and purposeful events. Due to their popularity and presence in almost all human cultures, they gather many different stakeholder groups [1,2]. It is worth noting that food festivals are perceived as tools for sustainable development. This concerns not only the context of food and agricultural policy but also social sustainability [3] or sustainable place development by increasing the attractiveness and competitiveness of these places [4]. In addition, food festivals give the opportunity to preserve the tradition, culture, and identity of the area based on local products, creating jobs (counteracting depopulation), as well as raising environmental awareness (small handicraft companies are characterized by limited emissions). A food festival, as part of the local gastronomy, can contribute to sustainable development by combining different activity areas, such as agricultural production, integration, education, nature conservation, or tourism It also has implications for agricultural food security [5–7].

Vendors are one of the characteristic stakeholder groups of food festivals. These are usually food producers running small family businesses or farms. They manufacture their products on a small scale, taking care of the quality of the ingredients and the recipe. The food festival is a unique (often the only) opportunity for the vendor to meet directly with customers and other vendors [3,5,8,9]. A vendor of a food festival is an entrepreneur producing and selling food products, operating in a specific environment. There are also other participants in this environment, and certain rules apply. Generally, in the literature, such an environment in which an entrepreneur operates is called the entrepreneurship

ecosystem. Due to the fact that entrepreneurial ecosystems are defined in many ways with different elements [10], the literature indicates the need to study the entrepreneurship model, taking into account a possibly comprehensive set of elements [11]. Entrepreneurial ecosystems are inherently complex, and knowledge regarding the interactions between the different components is still needed [10]. Questions regarding what actually constitutes an entrepreneurial ecosystem and how that ecosystem is managed continue to emerge as new and compelling issues [12].

It should be emphasized that the literature lacks a comprehensive approach to the entrepreneurial ecosystem in relation to food festivals. The research relates mainly to entrepreneurship and business in general on a specific territorial level, e.g., local or regional [13–18], within which the festival can operate [19]. In contrast, individual types of entrepreneurship have not been studied in detail, resulting in an overemphasis on generalizations of entrepreneurship [20]. In terms of the entrepreneurial ecosystem of festivals, including food festivals and food vendors, there are few examples of such research [21–23]. They refer mainly to the entrepreneurship itself and the vendors, ignoring the characteristics of other elements of the entrepreneurship ecosystem of the food festival. However, as reported by Mason and Brown [24], entrepreneurship ecosystems may be industry-specific and differ in characteristics among industries. It is worth noting that Getz [25–27] points to the need to conduct comprehensive research on festivals and events, treating the lack of such an approach as a research gap. Food festivals have many characteristics, such as the participation of food vendors, direct producer–consumer interactions, and competitions and cooperation among vendors. These features influence the formation of the entrepreneurial ecosystem.

The proposed research is an attempt to fill the research gap in terms of defining the characteristic elements of the entrepreneurship ecosystem of the food festival and the main relationships within the ecosystem. The contribution of the research includes the development of knowledge about the entrepreneurial ecosystem of the food festival, as well as the creation of a framework for further research, assuming that the entrepreneurial ecosystem is a good frame to understand food festivals.

The following research questions arise in the scope of the listed research gaps:

Q1. What features and elements determine the specificity of the entrepreneurial ecosystem of the food festival?

Q2. What is the importance of the elements of the entrepreneurial ecosystem of the food festival for the vendors' activity?

The main purpose of the research is to indicate the elements and relations of the entrepreneurship ecosystem of food festivals.

The article is organized as follows: Section 2 contains a literature review. Section 3 includes a theoretical background, and Section 4 presents the study context. Section 5 describes materials and methods. Section 6 presents the research results. Section 7 includes a discussion of the results, and Section 8 elaborates the study's conclusions.

## 2. Entrepreneurship Ecosystem—Literature Review

### 2.1. The Concept of Entrepreneurial Ecosystems

The concept of entrepreneurial ecosystems refers to the concept of an ecosystem in the understanding of natural sciences (generally defined as biotic elements, their physical environment, and all possible interactions among them). It is emphasized, however, that this analogy should not be taken too literally but rather as a metaphor [14,28–31]. Acs et al. [28] indicate that with regard to their genesis, entrepreneurial ecosystems are based on ecological systemic thinking, focusing on the interdependence of actors in a specific community in order to create new value.

Malecki [32] emphasizes that the concept of an entrepreneurial ecosystem is relatively new and emerged from various sources. As a result, there is no universally accepted definition of this term. However, the author points out that most definitions emphasize the connection or interaction of elements, often through networks, creating shared cultural

values that support entrepreneurial activities. A similar opinion is shared by Autio and Levie [33], emphasizing that the lack of coherent theoretical foundations is reflected in the diversity of definitions. Spigel and Harrison [34] also point to the lack of a solid theoretical foundation for business ecosystems.

Colombo and Dagnino [29] indicate that there are two different points of view on the evolution of entrepreneurial ecosystems: a bottom-up approach and a top-down approach. The bottom-up approach assumes that ecosystems evolve over time, just like natural ecosystems. The top-down approach suggests that ecosystems can even be created from scratch or shaped. Such ecosystems are created and evolve less over time. A similar observation was made by Cantner et al. [35]. The authors emphasize that the core of the ecosystem, the source of creation and change, is the decision-making process. Both how ideas are generated and used are important—either internally within established companies or by setting up a new venture as a way to commercialize an idea.

### 2.2. The Definition and Characteristic of Entrepreneurial Ecosystems

According to O'Connor et al. [36], key to the definition of entrepreneurial ecosystems are (entrepreneurial) agency and (human-made) context (the ecosystem), especially constraints invented by humans (e.g., rules and institutions). Indeed, these elements appear in different definitions.

Isenberg [37] gives a simple definition, according to which the entrepreneurship ecosystem consists of a set of individual elements—such as leadership, culture, capital markets, and openminded customers—that combine in complex ways. A synthetic version of the definition is also proposed by Bruns et al. [38]. According to the authors, the entrepreneurial ecosystem is a multidimensional set of interacting factors that mitigate the impact of entrepreneurial activity on economic growth.

Public support, including local government support, is also important [39]. Mason and Brown [24] emphasize the importance of the local environment and the formal and informal character of the relationship. The authors define the entrepreneurship ecosystem as a set of interconnected entrepreneurial actors (both potential and existing), entrepreneurial organizations, institutions, and entrepreneurial processes, which formally and informally coalesce to connect, mediate, and govern the performance within the local entrepreneurial environment. Audretsch and Belitski [40] also point to the formal and informal aspect of the relationship in the ecosystem, defining the entrepreneurial ecosystem as "a complex system of interactions between agents within various socioeconomic, institutional and informational contexts which generate more new businesses and growth".

A regional and supporting reference is contained in the Spigel [41] definition, according to which entrepreneurial ecosystems are a combination of social, political, and cultural elements within a region that support the development and growth of innovative startups and encourage nascent entrepreneurs and other actors to take the risks of starting, funding, and otherwise assisting high-risk ventures. Similarly, Stam and Spigel [30] refer to the territorial and supporting aspects, defining entrepreneurial ecosystems as a set of interdependent entities and factors coordinated in such a way as to enable productive entrepreneurship in a given territory. According to Spigel and Harrison [34], the concept of entrepreneurial ecosystems plays a protective role as the conceptual umbrella for the benefits and resources produced by a cohesive, typically regional, community of entrepreneurs and their supporters that help new high growth ventures form, survive, and expand. Daniel et al. [42] perceive the entrepreneurial ecosystem as a unique, diverse, complex phenomenon, both rooted in place and dependent on context and relative perspectives.

Undoubtedly, the presented definitions show that the entrepreneurship ecosystem is a phenomenon composed of many stakeholders, among whom there are relations and actions take place in a specific socio-economic environment. Therefore, the basis for the functioning of the entrepreneurship ecosystem lies in the relations among the stakeholders. Entrepreneurship ecosystem stakeholders are any entity that has an interest, actually or potentially, in supporting and encouraging more entrepreneurship in a specific geographic

region [43]. Mason and Brown [24] indicate key actors in entrepreneurial ecosystems: entrepreneurial actors (entrepreneurs), entrepreneurial resource providers (financial providers and networks), and entrepreneurial connectors (associations, communities, and centers).

The goals and activities of various stakeholders and entities in the entrepreneurial ecosystem differ, but they can also coincide to a large extent [33,43]. Simatupang et al. [43] emphasize the need to focus on sustainable and desirable results of the functioning entrepreneurial ecosystem. In this approach, the entrepreneurial ecosystem plays a sustainable role by balancing the activities and relations among stakeholders.

As emphasized by Stam and Spigel [30], entrepreneurial ecosystems are, by nature, a geographic perspective. This means that ecosystems focus on the cultures, institutions, and networks that form within a region over time rather than the emergence in global markets. The spatial nature of entrepreneurship ecosystems is also indicated by Schäfer [44]. According to the author, a perspective that loosens the relationship between entrepreneurial ecosystems and administrative areas should be considered. Entrepreneurial ecosystems are geographically limited but not limited to a specific geographic scale (e.g., city or region). These ecosystems may be industry-specific or may evolve from one industry to several industries [24]. Thus, entrepreneurial ecosystems differ in terms of participants, ventures, business models, supporting organizations, and cohesion around shared values and activities [45].

Cavallo et al. [14] relate one of the main advantages of the ecosystem concept. The use of this concept makes it clear that the previous linear model (e.g., value chain) has become obsolete because it does not address the multifaceted complexity of doing business. Brown and Mason [46] confirm that entrepreneurial ecosystems are a very diverse, multistakeholder phenomenon that require political intervention. So far, it has not been possible to fully understand the complexity of this phenomenon, either on the scientific or political level. Roundy [47] emphasizes that the complexity of entrepreneurship ecosystems results from the fact that such a system is both created by the activities of individual entrepreneurs and shapes the activities of these individual entities.

Szerb et al. [48] indicate that the entrepreneurial ecosystems approach includes environmental, ecosystem, and performance metrics. Audretsch et al. [49] indicate the need to include in the entrepreneurship ecosystem supporting regulatory institutional solutions and government programs, informal networks, and entrepreneurial culture. According to the authors, an environment with regulatory institutions reduces unproductive entrepreneurship. Fuentelsaz et al. [50] also emphasize the importance of institutions for the functioning of the entrepreneurship ecosystem but point to the differentiation of support for entrepreneurship depending on the stage of the enterprise's life cycle.

According to Ratten [51], entrepreneurial ecosystems are based primarily on the self-organization of members around the pursuit of entrepreneurship. Furthermore, the more stakeholders involved, the more sustainably the ecosystem works. In addition, in the field of sustainability, the concept of sustainable entrepreneurial ecosystems is available in the literature on the subject. Such ecosystems function to support innovation and the creation of new companies, where sustainable development is the link between entrepreneurial activities [44]. As pointed out by Volkman et al. [52], this concept requires more research attention.

### 2.3. The Model of Entrepreneurial Ecosystems

Even though the entrepreneurial ecosystem is a conceptual model or strategy [53], Alvedalena and Boschma [54] note the lack of a clear analytical framework in the literature, whereas according to de Villiers Scheeperset al. [55], a frameworks is relatively simple to use and helps to identify some gaps. However, according to Hayter et al. [56], most research focuses on the individual, university, and company levels of the entrepreneurial ecosystem. This results in a lack of reference to vertical links between phenomena at the micro level and results at the macro level.

However, it should be emphasized that various models of the entrepreneurial ecosystem can be found in the literature. Stam [57] distinguishes among the elements of the entrepreneurial ecosystem: framework conditions (formal institution, culture, physical infrastructure, and demand) and systemic conditions (networks, leadership, finance, talent, knowledge, and support services/intermediate services). Stam [58] also proposed measures of such an entrepreneurial ecosystem, along with possible sources of data. It is worth explaining that framework conditions include social and physical conditions that enable or limit interpersonal interactions. Systemic conditions are the core of the entrepreneurial ecosystem. The individual components and their interrelationships are critical to the success of an ecosystem [30].

Suresh and Ramraj [11] presented the entrepreneurial ecosystem as a model of support, including the following elements: moral support (provided by loved ones and the community), financial support, network support (including the activities of organizations associating entrepreneurs, as well as online social networking sites), government support, technological support (provided by incubation centers or educational institutions), market support (created by market opportunities), social support (awards from trade associations and acceptance of venture failure), and environmental support (availability of natural resources and climatic conditions).

Stam and van de Ven [59] indicate the essential concepts of entrepreneurial ecosystem: institutions and resources. Each concept contains different elements. Institutions contain formal institutions, culture, and networks. Resources cover physical infrastructure, finance, leadership, talent, knowledge, and demand. Stam and van de Ven [59] emphasize the importance of new value creation as a result of the entrepreneurial ecosystem, which consists of intermediate services and productive entrepreneurship. This is a reference to earlier studies by Stam and Spigel [30], in which entrepreneurial activity is the result of the entrepreneurial ecosystem. It is the process by which individuals create opportunities for innovation. This innovation ultimately leads to the creation of new value in society; thus, it is the ultimate outcome of the entrepreneurial ecosystem, whereas entrepreneurial activity is more of an indirect outcome of the system.

Isenberg [60] proposed a comprehensive model approach to the entrepreneurship ecosystem, distinguishing six domains of this entrepreneurship: politics, finance, culture, support, human capital, and markets. Each domain contains subgroups consisting of elements. Policy refers to leadership and government. The policy should create institutional foundations and support for entrepreneurship. Finance includes primarily broadly understood financial capital from various sources needed to run a business. In the field of entrepreneurship culture, Isenberg [60] distinguishes success stories and societal norms describing good practices and features. Entrepreneurship support includes infrastructure, support professions, and non-governmental institutions. Human capital refers to both labor and educational institutions, emphasizing the necessary skills already at the level of education. Markets include networks and early customers, connecting both with other entrepreneurs and customers. Maroufkhani et al. [61] extended the Isenberg model [60] by adding two more domains, namely industrial dynamics (customer preferences, competitive situation, and technology) and crowdsourcing (crowdsourcing routine tasks, crowdsourcing complex tasks, and crowdsourcing creative tasks).

Due to the complex nature of the entrepreneurial ecosystem, Wurth et al. [62] propose a transdisciplinary research program of entrepreneurial ecosystems divided into four research streams (context, structure, micro-foundations, and complex systems) and four cross sectional themes (methodologies and measurements, theory, critical research, and transdisciplinary research).

## 3. Theoretical Background

Referring to the literature review, a definition of the entrepreneurship ecosystem of food festivals was formulated. It is a phenomenon composed of many stakeholders of

food festivals, among whom there are relations, and the actions take place in a specific socio-economic environment.

Inspired by the Isenberg model [60], the entrepreneurship ecosystem of food festivals vendors model was developed (Figure 1). The proposed model was divided into three dimensions: capital, micro-environment, and macro-environment. Capital is the core of the vendor's entrepreneurial ecosystem. Capital includes the conditions of the enterprise resulting from its features, properties, and resources. Economic capital has been separated, which consists of financial, product, and raw material resources, as well as production infrastructure necessary for the production process. The human capital covering the vendor's knowledge, skills, and education, as well as connections with the labor market, were taken into account. Social capital is also an important sphere. In the case of food festivals, culinary traditions are particularly important, often used by vendors and food producers. Another aspect of social capital included in the model is partnership, creating a cooperation network. This increases the possibilities of achieving common goals and mutual support. It is worth adding that the capital dimension is directly related to the micro-environment. Part of the capital comes from the micro-environment. Economic capital in the micro-environment may take the form of external capital or raw materials produced by external suppliers. Human capital is associated with the micro-environment, among others, through sources of knowledge and skills other than their own. Social capital is related to the micro-environment mainly through the relationships that build up or do not build trust and cooperation.

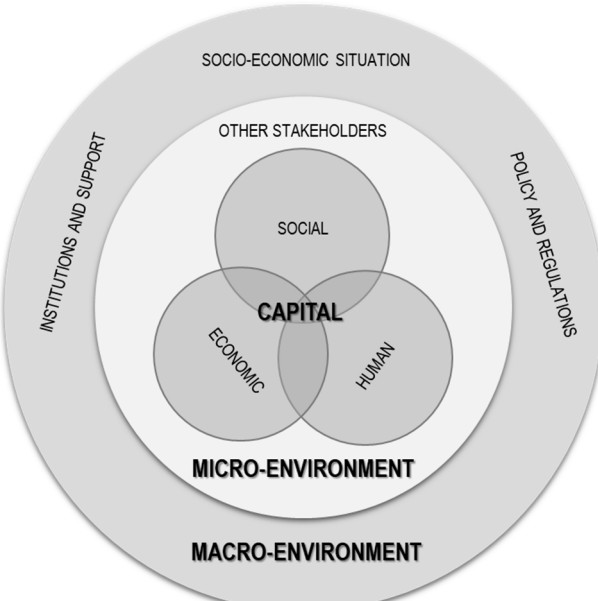

**Figure 1.** The entrepreneurship ecosystem model of food festivals—vendors approach. Source: Authors' own elaboration.

The structure and nature of the core is affected by the way that vendors conduct business. These are mainly small- and medium-sized enterprises, especially family businesses. In the case of family businesses, the activity is often related to the creation and preservation of values other than financial profits. Family-centered goals are important, non-financial goals, such as personal control, positive family image, sense of belonging. This contributes to the use of one's own family capital—both in terms of finances and labor resources, as well as tradition [63–65]. Generally, family capital is defined as total owning—family resources composed of human, social, and financial capital [66]. Hence, the first research proposition:

P1. The core of the entrepreneurship ecosystem model of food festivals is dominated by family capital (Figure 2).

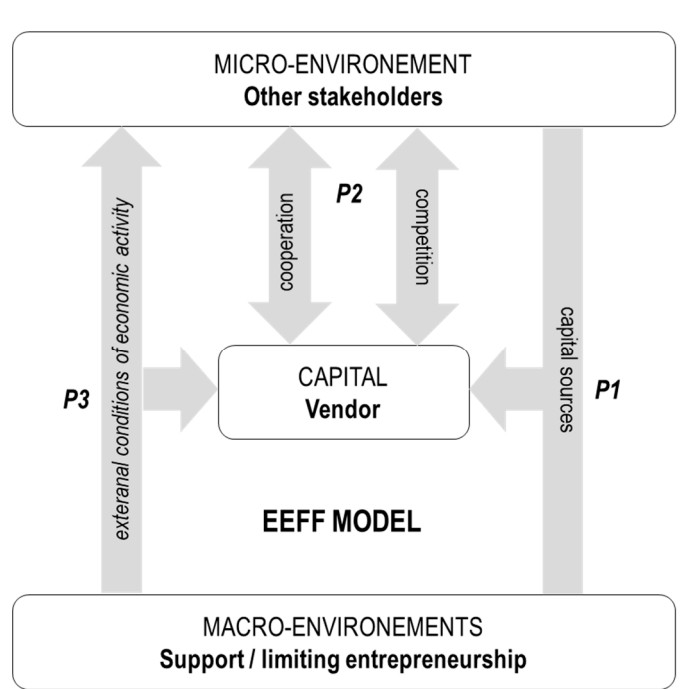

**Figure 2.** The entrepreneurship ecosystem model of food festivals—relations and propositions. Source: Authors' own elaboration.

The micro-environment of a food festival vendor is created by other stakeholders. Among others, these are customers entering into a purchase–sale relationship with the vendors. The micro-environment also includes other vendors—food producers who create competition but also provide opportunities for cooperation. The festival's organizers and other stakeholders also function in the micro-environment. The relations taking place in the micro-environment are direct. Their important feature is also the possibility of shaping them by the vendor. The micro-environment has the character of a closer environment, not because of the physical distance but because of the functional connections and the direct nature of the relationship.

The vendor has contact with other stakeholders, e.g., thanks to participation in the food festival. This gives an opportunity to meet a large group of people interested in culinary, both vendors and visitors/customers, as well as other stakeholders (organizers or sponsors). Thanks to this, food festivals give vendors the opportunity not only to sell products but also to exchange knowledge and experience [2,3,5,8]. Referring to the above points, the second research proposition was formulated:

P2. The food festival serves as a platform for connecting the vendor with the micro-environment (Figure 2).

The external dimension of the vendor's ecosystem is created by the macro-environment, the further environment. This is due to the indirect nature of the relations, as well as the vendor's inability to shape these relations. In the macro-environment, policy and regulations, institutions, and support, as well as the general socio-economic situation, should be distinguished. These are conditions beyond the control of the vendor but which affect the possibility of running a business. In order to be active, the vendor must adapt to the macro-environment.

The literature emphasizes the importance of formal institutions and regulations in the functioning of the ecosystem. Their main role is to define the basic conditions of economic activity, but they also affect the way entrepreneurship is run [11,30,47,50,59,60]. Institutions in crisis situations are particularly important. The event industry, of which food festivals are a part, needs institutional support, more so, as it is particularly susceptible to the impact of external factors, such as unforeseen crisis situations. This is due to their nature—they

gather a large group of people who interact directly. For visitors, they are primarily a form of spending free time, not a necessity; hence, in crisis situations, it is easy to resign from participating in the festival [67–69]. Taking into account the above statements, as well as the fact that the current times are unstable, the third research proposition was formulated:

P3. In the macro-environment of the food festival ecosystem, apart from the conditions of support, there are also factors that limit/hinder activities (Figure 2).

## 4. Study Context

Due to the diversity of food festivals in Poland, three different festivals were selected. The selection took into account the criterion of size, place of organization, duration, and subject matter. Diversity supports the comprehensiveness of the approach and helps to find answers to research questions. Therefore, the research was carried out at the Festival of Edible Flowers in Dobrzyca, the Festival of Good Taste in Poznań, and the Festival of Pomeranian Taste in Gdańsk. It should be emphasized that the research was carried out in 2020, during the COVID-19 pandemic, which made data collection difficult due to restrictions on participants and the number of vendors at selected festivals.

The Edible Flower Festival is organized every year in the summer. This festival takes place in a rural area and is organized by a family business that runs themed gardens and a gardening shop on a daily basis. The theme gardens are a well-known attraction in the area, and the company also organizes other events. The Edible Flower Festival is a small event organized in a rural area and associated with these areas. It promotes an alternative food trend, i.e., flowers, so it can be described as a monothematic food festival. It attracts primarily local and regional vendors, such as farmers, beekeepers, gardeners, and herbalists. These are mainly small food producers. They offer unique, high-quality products. This is a factor that attracts not only the local population but also tourists spending their holidays at the seaside (The festival is located a short distance from the Baltic Sea.) It is worth noting that the Festival is quite short—it is a weekend event with approximately 30 vendors.

The second festival selected for the study was the Festival of Good Taste in Poznań. It is a large festival organized in a large and popular city. The festival is not thematic, but its leitmotif is good, high-quality food. This festival combines culinary traditions with a modern approach to nutrition. Thus, it is a typical food festival with a variety of culinary themes. The vendors include both local and regional food producers and importers of food products from abroad. The Festival of Good Taste is well known to tourists—both Polish and foreign. Every year it attracts many visitors and many vendors. This festival can be found in tourist guides, as well as on the list of the ten most popular food festivals in Poland. The festival lasts four days, including the weekend. The festival is visited by approximately 50 vendors.

The third selected event is the Festival of Pomeranian Taste organized as part of the St. Dominic's Fair in Gdańsk. The St. Dominic's Fair is a well-known and respected event—both in Poland and abroad. It is the largest event of this type in Poland. It brings together many vendors from various industries, including food vendors. Therefore, the Fair is thematically diverse, but it also includes culinary offerings. The Festival of Pomeranian Taste is represented by regional producers of high-quality food, associated with the European Network of Regional Culinary Heritage. Therefore, the products on offer are dominated by those made on the basis of traditional recipes from regional ingredients. The range of offerings is quite wide and includes honey, dairy products, meats, fish, bread, fruit and vegetable preserves, oils, and alcohols. Due to the high quality of the products, they are popular among the visitors. The St. Dominic's Fair, including the Festival of Pomeranian Taste, lasts three weeks. The number of visitors at the Fair is huge, and the Festival of Pomeranian Taste has approximately 20 vendors.

## 5. Materials and Methods

The study adopted the qualitative approach based on the research design presented in the Figure 2. We tested the proposals for the culinary festival entrepreneurial ecosystem presented in Section 3:

P1. The core of the entrepreneurship ecosystem model of food festivals is dominated by family capital.

P2. The food festival serves as a platform for connecting the vendor with the micro-environment.

P3. In the macro-environment of the food festival ecosystem, apart from the conditions of support, there are also factors that limit/hinder activities.

Data were collected at three food festivals in Poland, in the summer (July–August) of 2020. The research took the form of semi-structured interviews [70]. Interviews were conducted with vendors. Questions and themes concerned individual elements of the entrepreneurship model of a food event: capital, micro-environment, and macro-environment. Model themes were addressed as open-ended questions, and the answers were noted. The interviews were not recorded due to the cultural and legal conditions prevailing in Poland in this respect. Each interview lasted approximately 30–40 min. This is a time-consuming method, but it allows one to create a fairly comprehensive picture of the situation. The research was conducted using the pen-and-paper interview method, which means that the researchers asked questions in person and recorded them on paper [71]. A group of researchers spent two days at each festival. The research team included three people directly related to the research and prepared from the substantive and technical side to conduct this research. This is an experienced group that has previously conducted this type of research. These are people professionally connected with science, as well as with the field of events.

In terms of sample selection, non-probability purposive sampling was used. This is a tool quite often used in the social sciences. Purposive sampling was selected, from among others, due to the qualitative nature of the research, the purpose of which is to deepen the understanding of the entrepreneurship ecosystem of the food festival. In addition, the conducted research is of a pilot nature, focusing on the initial diagnosis. Moreover, it is difficult to clearly estimate the number of local food producers participating in food festivals in Poland [72].

For these reasons, non-probability purposive sampling was used, consisting of the conscious selection of respondents due to their specific characteristics [73]. The research takes into account the following characteristics of the respondents: (1) a vendor at a food festival; (2) food producer; and (3) food produced on a small scale using local products. Fast food sellers as well as sellers offering non-food products were excluded from the study. Using these criteria, 58 full semi-structured interviews were obtained. According to the literature, there are no limits to how many respondents should constitute a sample in purposive sampling. In addition, despite the inherent bias, purposive sampling can provide reliable and robust data [72,74].

In order to determine whether the tested sample can be considered sufficient, the saturation of the data was checked [74]. The method proposed by Guest et al. [75] was used. According to this method, a base size from the first 4 interviews is determined, and individual new information is related to this value. The required information saturation threshold is 5%. According to the data in Table 1, the base size from the first 4 interviews produced 21 codes (out of 42 possible). Then, dividing the number of new codes in this series (4) by the number of unique codes in the base set yielded 19% new information. The last interview, characterized by the value of saturation with new information above the required 5%, was interview 19; in other cases, the values are below the required 5% threshold. This means that using base size 4, the threshold of 5% new information was reached with 19 interviews.

**Table 1.** Saturation assessment at base size 4 and run length 2.

| Interview Number | New Themes per IDI | New Themes in Run | % Change over Base |
|:---:|:---:|:---:|:---:|
| 1 | 14 | | |
| 2 | 4 | | |
| 3 | 2 | | |
| 4 | 1 | 21 | |
| 5 | 2 | | |
| 6 | 1 | 3 | 14.3 |
| 7 | 2 | 3 | 14.3 |
| 8 | 1 | 3 | 14.3 |
| 9 | 1 | 2 | 9.5 |
| 10 | 1 | 2 | 9.5 |
| 11 | 1 | 2 | 9.5 |
| 12 | 1 | 2 | 9.5 |
| 13 | 1 | 2 | 9.5 |
| 14 | 2 | 3 | 14.3 |
| 15 | 0 | 2 | 9.5 |
| 16 | 2 | 2 | 9.5 |
| 17 | 1 | 3 | 14.3 |
| 18 | 1 | 2 | 9.5 |
| 19 | 1 | 2 | 9.5 |

Source: Authors' own elaboration.

Due to the adopted research method, which is a semi-structured interview, qualitative data were obtained [76]. A coding technique was used to process the data [77], which made it possible to transform qualitative data into measurable data. The analysis was carried out in a deductive way. Interviews were conducted on the basis of parts of the theoretical model and related themes. The codes were specified separately for each taken up topic. Then, on the basis of the frequency of occurrence of the codes, the dominant category was established. Due to the qualitative nature of the study, the coding results were presented using code frequency in tables [77]. In terms of data presentation, an approach characteristic of qualitative research was used, where the results are presented under each main topic, additionally illustrated with literal quotations [78,79].

During the research, a total of 58 semi-structured interviews with food vendors were conducted. Due to the products offered by the vendors, several groups can be distinguished: honey and bee products, dairy products, alcohols, sweets, tinned food, bread and other baked goods, and cold cuts. The respondents were dominated by beekeepers, farmers, and small local food producers. Family businesses dominated; hence, a significant (25.8%) share of companies had been operating for more than 20 years. The average age of companies run by vendors was 17.7 years.

## 6. Results

Taking into account the economic capital (Table 2), the respondents were asked about production and the resource base. The interviews with vendors show that the majority (almost 90%) produce on their own the products they sell, based on their own production infrastructure. The surveyed vendors emphasized that with this type of production and scale, the infrastructure does not have to be developed and usually equipment that is located, for example, on a farm is sufficient. In terms of the raw material base, the respondents use mainly their own crops (62%). In addition to their own crops, raw materials from the local (29%), regional (19%), and national (17%) markets are used for production. It is worth noting that the domestic market comes from commonly used and standardized raw materials, for example, flour. Usually, the most unusual and non-standard ingredients come from the vendor's own production. Good quality raw materials, such as baking eggs, are often imported from the local and regional markets. The respondents emphasized that their own crops or own breeding guarantees the highest quality, proven raw materials, which affects the quality and unique nature of the products. Such an arrangement makes it

possible to sell one's own agricultural production at favorable prices. It is a particularly important source of income for small family farms. As one of the farmers said:

**Table 2.** Capital in the opinion of the respondents.

| Model Element | Theme | The Most Common Answers/Codes | % |
|---|---|---|---|
| Economic capital | Production | Self-Manufactured Products | 89.7 |
| | Resource base | Own crops | 62.1 |
| | | Local market | 29.3 |
| | | Regional market | 19.0 |
| | | National market | 17.2 |
| Human capital | Sources of knowledge and skills about products | Own knowledge, passed on in the family | 79.3 |
| | | Old and traditional recipes | 36.2 |
| | | Workshops and training | 31.0 |
| | Human resources | Permanent employment | 91.4 |
| | | Family members employment | 44.8 |
| | | Seasonal employment | 19.0 |
| Social capital | Non-business relations | Sharing passions and interests with others | 65.5 |
| | | Private contacts with other vendors | 39.7 |
| | | Private contacts with organizers | 13.8 |
| | Relationship with local and regional culinary traditions | The recipe of the offered products referring to local or regional traditions | 82.8 |
| | | Promoting culture and traditions about products and dishes | 65.5 |

Source: Authors' own elaboration.

"Agriculture is my passion and family tradition, and participation in festivals is my source of income and the opportunity to sell what I produce myself".

Sources of knowledge and skills harmonize with the raw material base. In this regard, producers also use their own potential as well as knowledge passed down from generation to generation (79%). The respondents also declared that they use old recipes (36%) as well as workshops and trainings to improve their skills in the production of products (31%). Continuing family traditions, using old, traditional recipes and enriching them with their own ideas proves the creativity of the respondents and fits with the trend of healthy eating, without preservatives, based on simple ingredients. Interestingly, some respondents experiment with new flavors or additions to their products, and others stay true to strictly developed recipes. Both tendencies have their supporters on the buyers' side.

Labor resources include mainly their own work, but more than 90% of respondents also have permanent employees. It is also worth noting that 45% of vendors employ family members. This means that vendors willingly use family work resources, explaining that such employees are better attached to the business, as well as being emotionally involved in the development of the company. The vendors also add that family workers usually do not need to be trained because they have been involved in the family business for a long time. They also enjoy more confidence. Less than 20% of vendors employ seasonal workers.

With regard to social capital, the research focused on non-business relations and references to local and regional culinary traditions. Respondents admitted that, in terms of non-business relations, the most important factor for them is the opportunity to share their passion with other people. In this regard, acquiring potential customers is not the most important aspect. For some respondents, the conducted production and vendor activity is not only a business but also a way of life. Usually, they successfully combine business with passion, and their experience and knowledge make them more credible to potential customers. They are happy to share their passion with everyone who wants to listen and try:

"It does not matter whether someone buys or not, the most important thing is the interest and the audience. If I am credible, customers will hear about me".

Some respondents maintain non-business contacts with other vendors (almost 40%). Most often, they share a common area of residence, they know each other as neighbors. This acquaintance often turns into help, e.g., they organize a journey together and a stand at the festival. Only 14% of respondents declared that they maintain non-business relations with the organizers. In this respect, business contacts related to festivals prevail.

An important element of social capital is the relationship with culture, traditions, and customs. Almost 83% of respondents declared that the recipes of the offered products refer to local or regional traditions. As already indicated, most of the knowledge regarding the products comes from family accounts or old traditional recipes. The respondents are aware that they are the link between customers and culinary traditions. Thus, 65.5% of respondents declared their willingness to disseminate culture and traditions regarding the products and dishes. This is an important aspect of food festivals—on the one hand, participants/viewers have the opportunity to try and buy traditional food, and on the other hand, vendors can present dishes or products that have been with their families for a long time and are associated with local traditions.

In terms of the micro-environment (Table 3), business relationships with other festival vendors focus on competition, collaboration, and customers. The analysis of the phenomenon of competition among vendors at the food festival provided interesting observations. Most of the respondents claim that their products are unrivaled—due to the originality of the recipe, quality of ingredients, and production. However, as the interviews show, the competitors observe each other, they know their assortment and prices, and often also their weaknesses and strengths. For 43% of respondents, the festival provides an opportunity to learn about the offerings of others. One-third of the respondents indicated the possibility of establishing contacts with competitors. Concerns about competition were declared by 28% of respondents. More than half of the respondents cooperate with other vendors, and the effect of this cooperation is primarily the expansion of the offerings and increased sales, as well as the exchange of experiences. The respondents also cooperate with the organizers outside the festivals (65.5%). These are typically business contacts. The vendors treat the festivals as a place where cooperation can also be established with farmers and other suppliers of resources (34.5%), as well as with intermediaries (17%).

**Table 3.** Micro-environment in the opinion of the respondents.

| Model Element | Theme | The Most Common Answers/Codes | % |
|---|---|---|---|
| Organizers | Cooperation with the organizers | Business contacts at the festival and also outside the festival | 65.5 |
| Other vendors | Other vendors as competition at the festival | The opportunity to see the offerings of other vendors | 43.1 |
| | | Possibility of establishing contacts with competitors | 31.0 |
| | | Concerns about competition | 27.6 |
| | Cooperation with other vendors | Collaboration in progress | 51.7 |
| | | The effect of cooperation—extending the offerings and increasing sales | 34.5 |
| | | The effect of cooperation—exchange of experiences | 22.4 |
| Customers | Sales range | National | 53.4 |
| | | Regional | 27.6 |
| | | Local | 19.0 |
| | Contact with customers at the festival | Selling products | 82.8 |
| | | Possibility to obtain opinions about products | 81.0 |
| | | Possibility of acquiring new customers | 63.8 |
| | Contact with customers outside the festival | Own stationary store | 51.7 |
| | | Products tastings | 51.7 |
| | | Own online store | 34.5 |
| | | Workshops | 34.5 |
| Other stakeholders | Possibility of establishing cooperation with other groups at the festival | With farmers and other resources suppliers | 41.4 |
| | | With intermediaries | 17.2 |

Source: Authors' own elaboration.

Customers are an important element of the micro-environment. Thanks to them, vendors gain income and can carry out their activities. For vendors, the festival is a place to sell products, and this is the main goal for the majority of respondents (83%), as well as the opportunity to obtain product opinions (81%). Some respondents further emphasized the possibility of acquiring new customers (64%). This is due to the fact that some of the vendors have a group of regular customers who also buy their products outside the festivals. This is possible through the vendor's own store, with almost 52% of respondents having their own stationary store, and 34.5% online. More than half of the respondents described the range of product sales as national, 28% as regional, and 19% as local. Relations with customers also help to establish tastings (51.7%) and workshops (34.5%) organized by the vendors. It is worth adding that one of the factors attracting buyers to the food festivals is the increase in nutritional awareness. Products made of high-quality ingredients are sought as an alternative to the market offer. According to one of the vendors:

"People today want to eat well, not just eat a lot".

It is worth emphasizing that the interviews conducted show that participants of food festivals create their unique social capital. Relationships are built, often informal, going beyond the time and place of the festival.

With regard to the macro-environment (Table 4), respondents were asked about difficulties in business activity, the use of support for entrepreneurs, and the impact of the COVID-19 pandemic on their activities. Among the difficulties resulting from regulations and policies, as well as from legal and administrative conditions, the respondents most often mentioned the complicated legal framework for running a business in Poland (41%). According to the respondents, these regulations should be simpler and easier to understand for entrepreneurs, so that they can focus on their activities and not on administrative and legal problems.

**Table 4.** Macro-environment in the opinion of the respondents.

| Model Element | Theme | The Most Common Answers/Codes | % |
|---|---|---|---|
| Policy and regulations | Business difficulties | Complicated legal framework | 41.4 |
| | | Frequent changes in regulations | 32.8 |
| | | Excessive taxation | 27.6 |
| Institutions and support | Using support for entrepreneurs | Fairly good integration of business with supporting institutions | 58.6 |
| | | Possibility of using aid programs for SMEs | 53.4 |
| | | Excessive bureaucracy | 29.3 |
| Socio-economic situation | Impact of the COVID-19 pandemic | Decrease/no sales of products at the festival | 62.1 |
| | | No impact on sales outside the festival | 37.9 |

According to the respondents, excessive complexity of regulations may even become a barrier to economic activity, especially while also adding another difficulty indicated by the vendors—frequent changes to the regulations (33%). This variability makes it necessary to constantly observe the regulations, as well as search for their interpretation. This is often connected with the necessity to take advantage of a specialist's advice (also for a fee). One of the respondents clearly emphasized:

"I am not a lawyer, I am a farmer, I know the land, but not the rules".

Many other opinions were similar. The respondents are experts in their field, but not necessarily in the field of law.

Another business limitation is excessive taxation. In the opinion of the respondents, it often results in insufficient funds remaining in their disposable budget. They must calculate the project well to make it economically viable. Producers perceive it as a significant limitation to the possibilities of their activity in the financial sense.

In terms of support for entrepreneurs, the respondents emphasized that the integration of business with supporting institutions is quite good (59%) and that there are many

possibilities of using such support (53%), but the main limitation is excessive bureaucracy (29%). Some respondents confirmed that they had used or still use various forms of support for small- and medium-sized enterprises. It is an important development factor for them. However, those respondents who did not use the support are aware that such a possibility exists.

Vendors were also asked about the impact of the COVID-19 pandemic on their business. The answers were varied. Overall, 62% of respondents recorded a drop in sales at festivals—mainly due to the reduction and cancellation of many events during this period. It is worth adding that 38% of producers did not notice a drop in sales outside the festivals. Some even saw an increase—especially producers of oils and honey. Some of these products are associated with increasing immunity and health. Pandemic restrictions were felt relatively less by vendors whose business is based on activities outside the festivals (e.g., farmers) and much stronger for vendors whose activities are based mainly on the sale of products during festivals.

## 7. Discussion

According to the views of Mason and Brown [24], the entrepreneurial ecosystem can be industry-specific. The article presents an ecosystem approach to a segment of the event industry—food festivals. The specificity of such an entrepreneurship ecosystem results from the characteristics of food festivals—specific capital forming the core of the ecosystem, stakeholder groups (organizers, vendors, and visitors) functioning in the micro-environment, as well as external conditions creating the macro-environment. A characteristic feature of the entrepreneurship ecosystem model of food festivals is also cohesion around common values and activities related to food, culinary heritage, which is in line with the views on entrepreneurship ecosystems [45]. The proposed entrepreneurship ecosystem model of food festivals was inspired by the research of Isenberg [60]. However, it takes into account the specificity of food festivals. Filling the research gap regarding the lack of defining the characteristic elements of the entrepreneurship ecosystem of the food festival, it was determined that such an ecosystem consists of three parts: capital, micro-environment, and macro-environment. The parts influence each other, causing relations among them and within them. This approach only goes beyond the entrepreneurial approach presented by Kwiatkowski et al. [21] and Hjalager and Kwiatkowski [22].

The research shows that vendors/food producers largely use their own capital (P1 confirmed). This applies to both economic and human capital. The activity is usually carried out on the basis of self-manufactured products using their own raw material base (own crops). A significant portion of vendors/producers employ family members or use family help. In most of the production, they also use their own knowledge—passed down in the family from generation to generation. Therefore, the basis of this type of activity is family capital, which is consistent with the results of research by Berrone et al. [63], Hauck [64], and Hasan [65].

According to our findings, for the vendors/producers, the food festival acts as a platform connecting with the micro-environment (P2 confirmed). For the surveyed vendors, the possibility of establishing business contacts with the organizers at the festivals and maintaining them also outside the festivals resulted in being particularly important. In addition, vendors emphasize that thanks to participation in the festivals, they can become acquainted with the offerings of others, as well as establish contacts and cooperation with other vendors, including competitors. In addition to the possibility of selling their products, vendors can form an opinion about their products from the target group. Therefore, food festivals offer many opportunities for vendors, which is in line with the results of other studies [2,3,5,8].

The entrepreneurship ecosystem model of food festivals is not a typical support model presented by Suresh and Ramraj [11]. According to our findings, in the macro-environment of the food festival ecosystem, apart from the conditions of support, there are also factors hindering activity (P3 confirmed). Most of the respondents benefit from institutional and

government support addressed to small- and medium-sized enterprises, which confirms the results of research on the importance of formal institutions [11,30,47,50,59,60]. It is worth emphasizing, however, that apart from institutional support, the respondents pointed to formal difficulties in running a business. The complicated legal framework, frequent changes in regulations, as well as excessive taxation, are pointed out. In addition, the vendors emphasize that their activities are largely influenced by external conditions, independent of them. Most of them negatively felt the effects of the COVID-19 pandemic, which is consistent with the research conducted by Janiszewska et al. [67], Dragin-Jensen et al. [69], and Kwiatkowski et al. [67]. The vendors emphasize the significant role of institutional and government support in such crisis situations.

It is worth emphasizing the observed influence of the food festival on sustainable development. The festivals connect various stakeholder groups by building relationships among them. These relationships often last much longer than the festival, contributing to building social sustainability. Our observations in this regard are consistent with the results of De Jong, Varley [3]. According to our observations, the vendors of food festivals support local tradition and culture, e.g., by using traditional recipes and producing local products. Thus, they combine the economic and social planes of sustainable development. The conducted research confirms the findings of Baldy [6]. In addition, the entrepreneurial ecosystem of the food festival refers to sustainability by its very definition. Combining different elements, different stakeholders, and different factors, it plays a sustainable role. Thus, it is a type of sustainable entrepreneurial ecosystem, which refers to the research of Simatupang et al. [43].

The limitations of the presented research result from two main reasons. Firstly, the research was conducted on a limited number of food festivals and vendors, which was caused by the COVID-19 pandemic. Secondly, the issue of entrepreneurship ecosystems is complex and its empirical research requires the use of simplifications. Research is limited to vendors and specific elements of the entrepreneurial ecosystem of food festivals.

The entrepreneurship ecosystem model of food festivals requires further research. Due to the holistic approach, additional analyses would be required for individual elements and other stakeholder groups of such an ecosystem. Another direction for further research should be the evolution of the entrepreneurship ecosystem of food festivals. The ecosystem is a dynamic phenomenon—both due to the multitude of stakeholders and factors and due to changes over time, e.g., as a result of the impact of external factors. It is also worth considering a broader approach to the sustainable entrepreneurship ecosystem of food festivals in future research.

## 8. Conclusions

The conducted research indicates the considerable complexity of the entrepreneurship ecosystem model of food festivals and the need for further research. The article attempts to identify the main elements of this ecosystem and the main relationships within it. This is an approach from the point of view of the food vendor/producer. The core of the ecosystem was distinguished, consisting of the capital of the food vendor/producer (economic, human, and social), the micro-environment, including the festival's stakeholders, and the macro-environment, taking into account policy and regulations, institutions and support, and the socio-economic situation.

The conducted research shows that the core of the entrepreneurship ecosystem model of food festivals is dominated by the equity of the food vendor/producer, referring to both economic and human capital. In addition, for the vendor/producer, the food festival acts as a platform connecting with the micro-environment. At the same time, a food festival is often the only opportunity for the vendor to have direct contact with customers or other vendors, including competitors. Despite the fact that one of the main roles of the entrepreneurship ecosystem is to support this entrepreneurship, in the macro-environment of the food festival ecosystem, apart from the conditions of support, there are also factors

that hinder the activity. These factors result not only from the general socio-economic situation but also from the regulations applicable to vendors.

The presented research is only an attempt to show how the entrepreneurship ecosystem model of food festivals works. They in no way exhaust the topics covered. The presented observations require further in-depth research, e.g., in terms of the evolution of this ecosystem or the dynamics of relationships.

**Author Contributions:** Conceptualization, L.O., D.J. and G.K.; methodology, L.O., D.J. and G.K.; software, L.O., D.J. and G.K.; validation, L.O., D.J. and G.K.; formal analysis, L.O., D.J. and G.K.; investigation, L.O., D.J. and G.K.; resources, L.O., D.J. and G.K.; data curation, L.O., D.J. and G.K.; writing—original draft preparation, L.O., D.J. and G.K.; writing—review and editing, L.O., D.J. and G.K.; visualization, L.O., D.J. and G.K.; supervision, L.O., D.J. and G.K.; project administration, L.O., D.J. and G.K.; funding acquisition, L.O., D.J. and G.K. All authors have read and agreed to the published version of the manuscript.

**Funding:** This research was funded by the National Science Center in Poland (grant number 2019/33/B/HS4/02068), and the APC was funded by the National Science Center in Poland (grant number 2019/33/B/HS4/02068).

**Institutional Review Board Statement:** Not applicable.

**Informed Consent Statement:** Not applicable.

**Data Availability Statement:** Not applicable.

**Conflicts of Interest:** The authors declare no conflict of interest.

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
