# Peer review of "The Entrepreneurship Ecosystem of Food Festivals—A Vendors’ Approach"

_sustainability, doi:10.3390/su15020906_

Round 1

Reviewer 1 Report

Honestly, the topic and the paper itself are extremely interesting to me. Entrepreneurial ecosystems are obviously a hot topic, a specific business situation is analyzed here. However, I would have a few comments that could affect the quality of the material:

1. The three hypotheses on which the research is based are stated, but they are very, I would say, clumsily stated in the part of the theoretical background. In my opinion, there should be a section on Methodology, where the Hypotheses come with other elements of the research setting, which are also given here, again, in several special parts.

2. Very shyly, I would say, it was stated that the hypotheses were proven, as well as the way in which they were proven and presented.

3. The display of results is based on basic statistics, with which you cannot prove anything, especially not research hypotheses. That's mostly where my decision towards moderate changes in the presentation came from - without this, the manuscript is not acceptable.

What cannot be changed, however, unfortunately, is the sample, which is extremely debatable. In the research, 58 respondents were interviewed - does this mean what was stated as "58 semi-structured interviews with food vendors were conducted"? If I am wrong, I apologize and I would like it to be so. However, if so, if this is true, then this survey is extremely unrepresentative or, at best, debatably representative. If you do not convince me of the representativeness of the sample, I intend to reject the paper in the following.

Reviewer 2 Report

I was pleased to read the study as it contains a very interesting approach by which the food festival is understood as entrepreneurial ecosystem. Overall, the paper is well organized, the literature review along with relevant theories is nicely conducted, the study context is clearly introduced, and the analysis is properly conducted. In sum, my impression about the paper is that it fits with the purpose of the special issue, “Sustainability of Festivals and Events.” My comments may be trivial but I hope the authors nicely consider them in the revised version. 

The authors have developed three hypotheses in the section “3. Theoretical background.” This section is very important because based on the previous section introducing models about entrepreneurial ecosystem,  it develops the author’s suggestion that entrepreneurial ecosystem is a good frame to understand food festivals. However, based on the conventions of scientific research, the sentences that the authors developed are better called “propositions” than “hypotheses” because I do not find any specific relationships between concepts in the hypotheses the authors introduced as they are. So, I would like to suggest that the current hypotheses are renamed as “propositions.”

The current H1 states “The core […]  is dominated by equity. (line 273-274)“ However, to me, two types of capital that the authors emphasize are not simply of equity: social capital and human capital. These two capitals are not actual equity-based capita, but metaphorical concepts. Given that, I doubt that the equity is a proper concept encompassing all economic, social, and human capitals that the authors introduce in the paper. I would like to ask the authors to find other proper concepts to include all these capitals, rather than equity. 

I found some typos and minor-level grammatical errors, so suggesting proper English proofreading in the further revision. 

Reviewer 3 Report

This is an interesting study, and the authors have collected a unique dataset using cutting edge methodology. The paper is generally well written and structured. The rising number of publications in this area indicates that it is an active area of research, so the results are quite interesting and make a strong contribution to the literature on this critical area of research. I am generally very positive about this article, but before publication, certain points should be revised, which are listed below:

The literature review should be more clear and specific. For this, you need to add some previous studies regarding the entrepreneurship ecosystem of food festivals. add these paper to your manuscript:

1.      Ashraf, U., Ashraf, A., Imran, M., Akhter, M. (2022). Impact of Climate Change on Agriculture Sector in Pakistan: A Case of District Lodhran, Southern Punjab-Pakistan Muhammad. Pakistan Journal of Life & Social Sciences20(1), 57-62.

2.     Gabinete, G., Tanan, C., Tutor, J. A., & Escantilla-Lebuna, M. L. (2022). Public Service Delivery Assessment Using the Citizen Satisfaction Index System in Western Visayas, Philippines. Pakistan Journal of Life & Social Sciences20(1), 36-44.

 Please consider adding "self of the-researcher" in the methods section. I believe that knowing the professional contexts of the researchers in relationship to the study and its participants gives us readers some important insights in the approach the researchers take to the study.

 Please elaborate your exclusion criteria.

Reviewer 4 Report

Dear Authors,

Thank you very much for this opportunity to read you work. The study aims to indicate the areas of the entrepreneurship ecosystem of food festivals and to determine their impact on the activities of food vendors. Overall, this is a well structured paper. On the other hand, several points should revised and addressed as follow: 

1. Abstract: Authors need fully to rewrite this section. This is the weakest section across the study. (1) The aim of the study is not clear and vague; (2) Methodology part is missing; (3) Results within the abstract section are not addressed well; (4) Also, the study contributions are not demonstrated well. 

2. Introduction: (1) Authors need to extend the study background relevant to its objectives; (2) Although the study gap was highlighted by authors, still the study objectives need to be rewritten again as they are not clear and vague. (3) In line 63;" In terms of the entrepreneurial ecosystem of festivals, including food festivals and food vendors, there are few examples of such research [20-22]" what are the gap found relevant to the study aim and how the authors intend to close the expected gap in those studies? 

3. Literature: Overall, this is a good section but authors need to link current and relevant studies to the study aim and to support their arguments in developing the study theoretical model. Further, authors need to draw the study model to show the cause-effect relationship among the study variables. 

4. Methodology: Based on the aim of the study, authors need strongly to justify the study design (major point). From my view, the study suffers  extremely from the selection of its design, method and findings generalization. Also, authors need to present the sampling technique used and how they analyze the data collected. Further information related to the study data collection methods are needed also. How authors assure the results reliability, validity and generalizability. How authors also deal with the qualitative design bias and sampling technique bias? 

Discussion: needs to be extend and link the study finding to the gap found in the literature. 

References: Justify the use of the following references: 

Ref; 20; 21; 64; 65; 66; 

Round 2

Reviewer 1 Report

The qualitative nature of the research cannot justify the absence of serious statistics and, to put it mildly, a very limited research sample. I consider this to be a serious shortcoming of the research as such, but I support the possibility that the mauscript will be accepted in this form as well.

Reviewer 2 Report

All my comments are well addressed in the revised manuscript.

Good job!

Reviewer 4 Report

Dear Authors,

Thanks a lot for working on the submitted draft. Overall, the draft looks fine now. However, try to address some points before the final publication stage:

1. Do note use any kind of abbreviation within the abstract section:  Please identify the Pen-and-Paper Personal Interview (PAPI).

2. Kindly mentioned that the study adopted the qualitative approach at the beginning of the methodology section along with the research design adopted. 

3. Kindly present the study philosophical stand also within the methodology section.

4. Please proofread your work as the paper suffers from several English grammar mistakes that should be checked and corrected.   

English language and style are fine/minor spell check required
